# Evaluating the effect of the SMART intervention in people with recently diagnosed breast cancer who are being treated at a public tertiary hospital in Australia: protocol and statistical analysis plan for a single-blinded, single centre randomised controlled trial

Susan Stinton[1]*, Dale Edwick[2], Chloe Maxwell-Smith[3], Natasha Bear[4], Lauren J. Breen[3], Alejandro Dominguez Garcia[3], Elizabeth Dylke[5], Kate Edwards[5], Sally Lynch[1], Adam Lloyd[6], Barbara Mullan[3], Niamh Moloney[2], Ru-Wen Teh[7], Carol Watson[1], Kylie Hill[1,2]

**1** Physiotherapy Department, Royal Perth Hospital, East Metropolitan Health Service, Perth, Western Australia, Australia, **2** Curtin School of Allied Health, Curtin University, Perth, Western Australia, Australia, **3** School of Population Health, Curtin University, Perth, Western Australia, Australia, **4** Institute for Health Research, Notre Dame University, Perth, Western Australia, Australia, **5** Faculty of Medicine and Health, The University of Sydney, New South Wales, Australia, **6** Community & Virtual Care, East Metropolitan Health Service, Perth, Western Australia, Australia, **7** Medical Oncology Department, Royal Perth Hospital, Perth, Western Australia, Australia

☉ These authors contributed equally to this work.
* susan.stinton2@health.wa.gov.au

## Abstract

### Introduction

Adults undergoing treatment for breast cancer (BC) are advised to participate in regular exercise. However, many struggle to exercise consistently due to the side effects of systemic treatments including nausea, fatigue, and pain. In adults with newly diagnosed BC, this trial will evaluate the effectiveness of a new exercise intervention, compared with usual care, on outcomes including health-related quality of life (HRQoL).

### Materials

This randomised controlled trial is underway at an Australian tertiary hospital. The protocol was prospectively registered (Australian New Zealand Clinical Trials RegistryACTRN12623001168640p). Consenting adults with BC diagnosed within the prior six months, with planned chemotherapy and/or endocrine treatment will be randomised to an intervention or control group. Both groups receive usual physiotherapy and medical care. Those allocated to the intervention group are offered participation in the 'SMART' intervention (Self-determined, Monitored, Adaptable, Rehabilitation with Telehealth support). This involves 16-weeks of tailored, one-on-one

**Data availability statement:** No datasets were generated or analysed during the current study. All relevant data from this study will be made available upon study completion.

**Funding:** This research is funded by the Vonesch Breast Cancer Rehabilitation Grant (S01/2023), through the RPH Research Foundation. This funding was made possible from a gift in Memory of Felix Viktor Vonesch and Verena Vonesch. Funding acquisition: KH, AL, CW The RPH Research Foundation have provided peer review of this protocol design but do not play a role in the conduct of the study; collection, management, analysis, and interpretation of the data; preparation, review or approval of the manuscript; and decision to submit the manuscript for publication. https://www.rphresearchfoundation.org.au/.

**Competing interests:** The authors have declared that no competing interests exist.

physiotherapy-led exercise sessions including behaviour change techniques and the weekly goal of completing 150 minutes of aerobic exercise and two resistance training sessions. The primary outcome is HRQoL and secondary outcomes include physical assessments (muscle strength, exercise tolerance, body composition), healthcare utilisation, workplace absenteeism, mood, psychological determinants of behaviour change, chemotherapy completion rates and endocrine therapy completion. All outcomes are measured prior to randomisation and 16 weeks following randomisation. Additional assessments of all outcomes (excluding the physical assessment) occur at 8 weeks and 52 weeks following randomisation. Ongoing recruitment for two years from June 2024 is expected to achieve a sample size of 260. No results have been analysed.

## Discussion

If the SMART intervention produces favourable change, this will support its adoption in clinical practice. A greater understanding of factors including BC stage, treatment type or variables relating to the exercise program, that influence the magnitude of exercise-induced change on HRQoL will inform future exercise programs.

## Introduction

Breast cancer affects mainly cisgendered women with a global incidence of 2.26 million in 2020 [1]. Fortunately, breast cancer survival rates have improved with up to nine out of ten people diagnosed with breast cancer deemed to be cancer-free at five years following their initial diagnosis [2]. Although largely successful in terms of survival, systemic treatments including chemotherapy and endocrine treatments for breast cancer have many unwanted side effects, such as fatigue, pain and worsened mental health, which reduce health-related quality of life (HRQoL) for many years after completion of treatment [3]. Strategies are needed to improve HRQoL during and following completion of treatment [3].

National and international oncology societies recommend that people undergoing treatment for cancer engage in 150 minutes of moderate intensity aerobic exercise and two sessions of resistance exercise each week [4,5]. This recommendation is informed by systematic reviews and meta-analyses of randomised controlled trials (RCTs) that favour exercise groups over control groups for measures of HRQoL [6,7], pain [8], fatigue [9], physical function [10], and mood [11]. Preliminary data suggest that exercise undertaken during treatment for cancer may improve chemotherapy completion rates [12] and reduce breast cancer related mortality [11].

Notwithstanding these benefits, earlier work has reported that up to 78% of people undergoing treatment for breast cancer do not meet the recommended amount of exercise and further, 23% reduce their participation in exercise during cancer treatment [13]. Engaging in regular exercise during systemic treatments, such as chemotherapy and endocrine therapy, is challenging due to common and debilitating side effects such as nausea, fatigue and pain [14,15]. During this time, an intervention to

increase participation in exercise should aim to adopt a flexible and individualised approach, be endorsed and supported by oncology care teams, and include specific behaviour change techniques to assist individuals receiving treatment for breast cancer with navigating barriers [16].

To optimise exercise among adults undergoing treatment for breast cancer, an exercise intervention was co-designed by a team that included physiotherapists, oncologists, psychologists, exercise physiologists and consumer representatives with lived experience of breast cancer. This exercise intervention has several novel aspects such as; i) bespoke tailoring of the program which considers age, baseline fitness levels, individual preferences and breast cancer stage, ii) inclusion and detailed mapping of behaviour change techniques, and iii) close monitoring using wearable devices to allow modification of programs dependent on treatment and other relevant health symptoms.

The primary research question that we will answer is:

In adults who are newly diagnosed with breast cancer, does the SMART intervention, compared with usual care, change HRQoL (primary outcome), peripheral muscle force-generating capacity, body composition, exercise tolerance, exercise adherence, healthcare utilisation, workplace absenteeism and presenteeism, chemotherapy or endocrine completion rates (secondary outcomes) at the primary trial endpoint, 16 weeks following randomisation? Exploration of the secondary outcomes will be hypothesis generating and look for evidence of an effect in outcomes deemed relevant by a project steering committee. In addition, we plan to utilise this dataset to undertake a number of secondary analysis. These will also be hypothesis-generating and help to inform future work in this area. These secondary analysis include:

1. An exploration of variables that influence the magnitude of any change in HRQoL following completion of the SMART intervention. These potential moderators include treatment pathways, cancer stage, and the type, quantity and frequency of exercise undertaken. Understanding which, if any variables related to the exercise prescription are associated with an increased magnitude of change in outcomes will assist with optimising future exercise interventions.

2. To explore of any differences in the characteristics of those who respond well to exercise training (i.e., responders) versus those show minimal change (i.e., non-responders) will help with the design of future research that aims to improve outcomes in non-responders.

3. An exploration of the influence psychological determinants of behaviour change, such as capability, opportunity, and motivation, have on changes in physical activity levels from baseline to post-intervention and to long-term follow-up. This will help identify which behavioural determinants are most strongly associated with short-term improvements and long-term maintenance of physical activity.

4. An exploration of the associations between physical activity and psychological wellbeing, specifically changes in positive and negative affect, and impaired functioning due to mental health across the intervention and follow-up period. These analyses will help identify whether higher levels of physical activity are associated with improvements in psychological wellbeing and functioning.

## Methods

This RCT will evaluate the effect of an experimental exercise program called the SMART intervention (**S**elf-determined, **M**onitored, **A**daptable **R**ehabilitation program with **T**elehealth support) in adults who have been diagnosed with breast cancer within the previous 6 months who require chemotherapy and/or endocrine treatment. This study was registered with the Australian New Zealand Clinical Trials Registry (ACTRN12623001168640p) prior to commencing recruitment (detailed in *Supplementary Item 1*). Ethics approval has been given from the Central WA Health Human Research Ethics Committee (formally under Royal Perth Hospital Human Research Ethics Committee RGS00006136) and Curtin University Human Research Ethics Committee (HRE2024−0153). Site specific approvals were obtained from the participating health service. Any protocol (*Supplementary Item 2: Full protocol V3.0*) amendments will be submitted to this registry and

 

the ethics committees. This manuscript has been described in accordance with the SPIRIT guidelines (*SPIRIT:* http://spir-it-statement.org, checklist in *Supplementary Item 3).*

## Setting

This is a single-blinded randomised controlled trial that is being conducted at Royal Perth Hospital (RPH); a 450-bed tertiary public hospital in Australia. Royal Perth Hospital offers a multi-disciplinary breast cancer treatment clinic (RPH-BC) which treats people diagnosed with breast cancer who reside in metropolitan and rural locations throughout the state of Western Australia.

## Recruitment

Adults are eligible for inclusion if they have a primary breast cancer, diagnosed within the previous six months and have planned (or have started) chemotherapy or endocrine treatment for their breast cancer. Potential participants are excluded if they; i) are undergoing active treatment for any other primary cancer diagnosis or are receiving systemic treatment for a prior breast cancer diagnosis, ii) were diagnosed with breast cancer more than six months prior to recruitment or, iii) have a co-morbid condition which precludes independent participation in exercise. Potential participants with Stage 4 Breast Cancer (according to the American Joint Committee on Cancer Staging Manual, 8th Edition) [17] are eligible to participate in the study, providing they have consent from their treating medical oncologist. Participants are required to have telephone and internet access or agree to attend RPH for all assessments and relevant components of the SMART intervention.

Eligible participants are identified at the RPH-BC. A clinician who is independent to the study team makes first contact with potential participants. Those who meet the study criteria and express an interest in participating in the study are contacted by a member of the research team to provide verbal and written information about the study (*Supplementary Item 4*). The study is also advertised in hospital outpatient clinics using posters and pamphlets and potential participants are encouraged to contact the study team (via phone or email) to express interest.

## Consent and randomisation

Once the participant has had time to consider their enrolment, written informed consent is obtained prior to any data collection. The written informed consent form will be signed by the participant, the study researcher, and witnessed by an adult independent to the research team. Participants are required to complete all baseline assessments before proceeding to randomisation. Participants are randomised to the intervention group (IG) or the control group (CG) on a 1:1 ratio, according to a computer-generated randomisation sequence. This sequence is stratified for; i) treatment pathway (chemotherapy vs endocrine treatment vs combined chemotherapy and endocrine treatment) and, ii) whether the participant regularly engages in exercise (defined as participating in exercise at least once a month for the six months prior to recruitment) prior to their breast cancer diagnosis. The sequence is stored and concealed using a secure web-based data management application, Research Electronic Data Capture (REDCap). Fig 1 summarises the participant flow according to the *SPIRIT* schedule of enrolment, interventions, and assessments.

## Interventions

Participants in both groups receive usual care from the RPH-BC and oncology service. Specifically, all people who require mastectomy or surgical resection of tumours with sentinel node clearance and/or axillary clearance are referred to a physiotherapist. Pre- and post-operative physiotherapy care may include education regarding the importance of exercise throughout cancer treatment, post-operative upper limb range of motion exercises, and/or referral to off-site oncology exercise groups. Lymphoedema surveillance and management, lymphatic cording, reduced shoulder range of motion, and/or other post operative complications are managed by physiotherapists on an as-needed basis. In addition, people

| STUDY PERIOD | Enrolment | Allocation | Post-allocation | | | | Close-out |
|---|---|---|---|---|---|---|---|
| TIMEPOINT | *Baseline* | *0* | *Week 4* | *Week 8* | *Week 12* | *Week 16* | *1 year* |
| **ENROLMENT:** | | | | | | | |
| **Eligibility screen** | X | | | | | | |
| **Informed consent** | X | | | | | | |
| **Allocation** | | X | | | | | |
| **INTERVENTIONS:** | | | | | | | |
| **Control group [Usual care]** | | ◆————————————◆ | | | | | |
| **Intervention group** | | ◆————————————◆ | | | | | |
| **ASSESSMENTS:*** | | | | | | | |
| **Demographics** [Age, sex, postcode, ethnicity] | X | | | | | | |
| **Exercise uptake** [IPAQ] | X | | X | X | X | X | X |
| **Quality of life questionnaires** [EORTC QLQ C30 & BR 45, EQ5D5L] | X | | | X | | X | X |
| **Physical assessments** [Body composition, 6MWT, strength] | X | | | X | | X | X |
| **Psychological determinants of behavioural change questionnaires** [Intention to exercise, COM-B, BSCS, BSES, TSES, HTQ, SRBAI] | X | | | X | | X | X |
| **Mental health questionnaires** [PANAS, WSAS] | X | | | X | | X | X |
| **Healthcare utilisation** [iMTA MCQ] | | | | X | | X | X |
| **Workplace absenteeism** [iMTA PCQ] | | | | X | | X | X |
| **Medical information** [Cancer stage/type, surgery date/types, chemotherapy completion rates, endocrine treatment plan/deviations] | | | | | | X | X |
| **Acceptability/feasibility** [Acceptability/Feasibility questionnaire, exit interviews] | | | | | | X | |

*EORTC QLQ C30 & BR45: European Organisation for research and treatment of cancer quality of life core questionnaire and the breast cancer specific module, EQ5D5L: Euroqol 5D5L quality of life questionnaire, 6MWT: Six minute walk test, IPAQ: International Physical Activity Questionnaire, COM-B: Capability, Opportunity, Motivation – Behaviour, BSCS: Brief Self-Control Scale, BSES; Barrier Self-Efficacy Scale , TSES: Task Self-Efficacy Scale, HTQ: Habitual Tendencies Questionnaire, SRBAI: Self-Report Behavioural Automaticity Index, PANAS: Positive and Negative Affect Scale, WSAS: Work and Social Adjustment Scale.

**Fig 1. SPIRIT study protocol schedule of enrolment, interventions, and assessments.**

can request to see a physiotherapist and the medical team can refer people to a physiotherapist for education regarding participation in exercise while undergoing chemotherapy at RPH. Any concomitant care relating to breast cancer treatment and/or other medical conditions will be offered in both groups, as per usual care.

Those allocated to the IG will receive the SMART intervention, which was co-designed with people who have breast cancer, to address many of the known barriers to exercising during cancer treatment. Consumer representatives have collaborated on the development of the protocol from trial inception. The core components of the program include; i) up to 16 one-on-one training sessions over the 16-week intervention period, ii) access to semi-supervised group gym sessions for participants to carry out their individual exercise programs, and iii) monitoring of supervised and home exercise sessions using strategies (described below). The SMART program will be delivered in an adaptable way, specific to each person, according to treatment symptoms, and progressed/regressed accordingly. An example of a supervised exercise session is detailed in Table 1 below.

The guiding principles that have shaped this intervention are as follows:

*S*elf-determined: Participants are encouraged to select exercises they enjoy (e.g., walking, swimming, cycling, dance) at the time of day that suits them best. They are encouraged to exercise at home, in the community or can access the gymnasium in the Physiotherapy Department at set times each week. Participants are encouraged to use a diary to record participation in exercise and to rate their physical and emotional health on a Visual Analogue Scale (VAS). These entries are reviewed at their one-on-one sessions.

*M*onitored: Participants are encouraged to attend regular one-on-one 60-minute exercise training sessions with a physiotherapist. During these sessions, together with a physiotherapist, the participant will; i) reflect on the achievement of the target behaviour and their response to the exercise (e.g., pulse rate and rating of perceived exertion [RPE]) [18] over the previous week, ii) report any adverse events or post-exertional malaise, iii) undertakes aerobic and resistance exercise with feedback, adjustment and reassurance as needed, iv) plan the exercise training sessions for the following week, and v) discuss behaviour change techniques (BCTs) to optimise success with achieving the intervention message. [19] Physiotherapists providing the SMART intervention will be upskilled in these techniques through a custom written resource for the study and one-on-one education with the trial staff member supporting the psychological and behaviour change components [20]. Example BCTs include goal setting around the target behaviour, feedback from clinic physiotherapists, action- and coping-planning for managing barriers, and environmental restructuring.

*A*daptable: The one-on-one sessions may be scheduled weekly on a regular day and time, or if the participant is suffering from treatment side effects (e.g., nausea and vomiting from day two to seven following chemotherapy), scheduling of these appointments may be more variable. They can choose to complete these sessions in person (at RPH) or via a telehealth appointment.

*R*ehabilitation: The target behaviours of the SMART program are informed by guidelines endorsed by National and International organisation including the American College of Sports Medicine (ACSM) and Clinical Oncology Society of Australia (COSA) [5,21]. Participants are encouraged to engage in both moderate intensity aerobic and resistance exercise. Aerobic exercise has been defined as any exercise that involves continuous repetitive movement using large muscle groups or whole-body exercise. Moderate intensity has been defined as that which requires either; i) a heart rate at 60–75% of their age-predicted maximum heart rate [22] or, ii) effort that is perceived to be 'somewhat hard' or 13–14 on the Borg RPE [18]. Vigorous intensity aerobic exercise will be defined as exercise performed at over 75% of their age predicted maximum heart rate or a Borg RPE rating of 15 ('hard') or more. Resistance exercises are prescribed for both upper body and lower body muscle groups. The mode (i.e., free weights, TheraBand, body weight, machine based) and specific exercise are tailored to each participant, according to their preference. In order to prescribe the load used for each exercise, the 10 repetition-maximum (RM) is determined [23] and training is prescribed at this load for 2 sets of 8 repetitions. Exercises are progressed by increasing repetitions (to 12), then sets (to 3) then weight. Exercises will be progressed if the Borg RPE scale for each exercise is less than 13 OR if participants are able to complete more than 3

**Table 1. Example of a supervised exercise session. Note that all exercise prescription is based on individual preferences, abilities, medical considerations and goals.**

| Participant details | 62 year old female; right-sided breast cancer diagnosed five months ago.<br>Treatment to date:<br>1) 14 weeks post right mastectomy and sentinel node biopsy.<br>2) Hormone medication: letrozole commenced one month post operatively<br>Relevant history:<br>Mild right shoulder pain post operatively with overhead tasks, improving last two weeks. No other exercise precautions to note. Has never engaged in any resistance style exercises, enjoys walking regularly to keep active (usually manages one or two 40-minute walks per week, but hasn't returned to regular walking routine since her surgery. |
|---|---|
| Participant goals | 1. Short term: To return to usual 40 minute walks (3–4 km), 2–3 times per week, within the next 2 months, so that she can feel confident keeping up with 3 yr old grandchild at the park on the days she cares for them.<br>2. Long term goal: to implement and learn how to complete a whole body resistance exercise routine 2 times per week to do all she can to maintain her bone health, given she has started letrozole and is concerned that she may be at risk of osteoporosis in later life. |

Supervised exercise in session: Week 5 of SMART program

| Phase of exercise | Time taken | Details |
|---|---|---|
| Warm up | 5 minutes | 2 mins light walk on treadmill or bike prior to increasing intensity for moderate intensity aerobic exercise |
| Aerobic exercise:<br><br>Treadmill or bike | 20 minutes | Aim for moderate intensity (monitored via patient rated of 13, and/or pulse rate of 103–129 beats per minute) |
| Resistance:<br>2x upper body exercises<br>2x lower body exercises<br>2x core/back exercises | 25-30 minutes | 1) Bicep curls using 3 kg weight (10 repetition max (RM) load), 8 repetitions, 2–3 sets<br>2) Lateral raises using 1.5 kg weight (12RM load, lower load to ensure pain free following post operative shoulder pain), 10 repetitions, 2–3 sets<br>3) Leg press machine: 50 kg (10RM load), 8 repetitions, 2–3 sets<br>4) Knee extension machine: 20 kg (10RM load), 8 repetitions, 2–3 sets<br>5) Lat pull down machine: 12 kg (12RM load, lower load to ensure pain free following post operative shoulder pain), 10 repetitions, 2–3 sets<br>6) Core exercise in lying with hips and knees at 90F with alternating heel taps: 8 repetitions each side, 2 sets |
| Cool down:<br>Stretches | 5 minutes | Lumbar spine rotation in supine (6 repetitions each side)<br>Shoulder elevation in supine (6 repetitions each side)<br>Thoracic openings in supine (6 repetitions each side)<br>Roll downs (6 repetitions) |

Home program

| Type of exercise | Time | Details |
|---|---|---|
| Aerobic | 20-30 minutes | Walking, at a moderate intensity (RPE 12–13) twice before next gym session. Walk times scheduled in both patient and her husband's shared diary with reminder on phone night before. Noted to meet longer term goal will need to increase to 3x 40–45 minute walks over the next 6–8 weeks. |
| Resistance | 20 minutes | Aim to complete once at home before next gym session<br>1) Bicep curls using blue TheraBand, 2 sets of 8 repetitions<br>2) Lateral raises (elbows flexed) to 90 abduction using green TheraBand, 2 sets of 10 repetitions<br>3) Squats holding shopping bags each with 3 kg weight (water bottles or similar), 2 sets of 8 repetitions<br>4) Bridging with single leg lift (alternating legs), 2–3 sets of 8 repetitions<br>5) Low rows using blue TheraBand, 3 sets of 8 repetitions<br>6) Core exercise in lying with legs at 90F with alternating heel taps: 6–8 repetitions each side, 2 sets<br>TheraBand set provided in session, exercises sent via PhysiTrack, bands to be set up at exercise station at home as environmental prompt. To check any post exercise symptoms over the coming weeks prior to progressing. |

sets of 12 repetitions without concern. Exercises will be progressed or regressed at the one-on-one sessions. If a participant has Stage 4 Breast Cancer, the exercise prescriptions set out above are adapted according to their individual safety considerations based on the location of metastases. [21]

*T*elehealth support: Participants are offered to attend their one-on-one sessions via telehealth. In this way, the study can enrol people who live in rural or remote areas of Western Australia. Additional online support using an online exercise prescription service, Physitrack, an online exercise prescription tool with a website or app interface (Physitrack PLC), will provide means for written exercise prescription, monitoring and exercise reminders.

## Managing hospital admissions, complications and sub-optimal adherence

It is anticipated that some study participants will require planned surgery such as the resection of their cancer, breast reconstruction procedures, and/or inpatient hospital care during the 16-week intervention period. Participants in both groups will have their 16-week intervention period paused during any inpatient admission or period of post-operative exercise restriction. For participants who develop medical complications such as low platelets, anaemia, or post-operative complications, exercise prescription will be modified following consultation with the treating oncologist and in accordance with national guidelines and position statements. [21,24]

In cases where participants in the IG do not report reasonable progress towards meeting their intervention goals over two consecutive weeks, the physiotherapist will meet with a member of the research team with expertise in the behaviour change theory to debrief and adjust the BCTs delivered if required. An example of an exercise plan is included in Table 2.

## Implementation fidelity

Fidelity with the SMART intervention will be assessed using two methods: participant reporting and wearable technology.

**Participant reports.** Participants in the IG are encouraged to diarise any participation in exercise as well as physical and emotional health on a VAS. These entries are reviewed at the one-on-one sessions. Content of these sessions will be documented by the physiotherapists. Additionally, participants will be contacted by the physiotherapist (via telephone) to complete the International Physical Activity Questionnaire (IPAQ) [25] at baseline assessment, on commencement of the intervention and at four-week intervals until completion of the intervention.

**Wearable technology.** For the duration of the intervention period, participants in the IG will be asked to wear a Fitbit (*Charge 5, Google Fitbit*) which records pulse rate and bodily movement (e.g., activity and exercise minutes). Data from the Fitbit will be compared with diary entries to corroborate participation in exercise [26].

**Table 2. Example exercise plan.**

| My exercise plan for week beginning 03/06/24: *Don't forget to log your exercises in your study booklet* | | |
| --- | --- | --- |
| Monday | 2pm | Gym session 60 mins (20–30 mins aerobic, 30 mins resistance exercises). Refer to your online exercise prescription. |
| Tuesday | 10am | Walk dog for 45 minutes (aim for heart rate of 120bpm) |
| Wednesday | 8am 1pm | Walk dog for 45 minutes (aim for heart rate of 120bpm) Chemotherapy appointment |
| Thursday | 10am | Gym session 60 mins (20–30 mins aerobic, 30 mins resistance exercises). Refer to your online exercise prescription. |
| Friday | | Anticipated day of rest following chemo |
| Saturday | | Anticipated day of rest following chemo |
| Sunday | | Day of rest or consider one of your low intensity programs (30 min light walk aiming for heart rate of 90bpm or your online yoga program), if you are starting to feel better after your chemo this week. |

## Measurement and blinding

At the time informed consent is obtained, participant characteristics (e.g., gender, height, weight, employment status, ethnicity, and postcode), breast cancer staging (according to the 8th Edition AJCC Manual) [17] and expected treatment pathway for their breast cancer (e.g., chemotherapy and/or endocrine therapy) will be recorded. If a participant requests to withdraw from the study, they will be asked if we can retain any previously collected data to be used in the analysis.

To reduce potential bias the person collecting outcome measures is blinded to group allocation. This is achieved through two strategies. First, baseline assessments will be completed prior to randomisation. Second, the physiotherapy staff who deliver usual care and/or the experimental intervention will not perform any follow-up assessments. Data collected via self-reported questionnaires will be completed by the participants via the REDCap automated survey invitation feature.

## Outcomes

Health-related quality-of-life is the primary outcome and will be assessed using three self-reported questionnaires; the disease specific European Organisation for Research and Treatment of Cancer (EORTC) Quality of Life C30 [27] questionnaire together with the breast cancer specific BR45 module [28], and the non-disease specific Euroqol EQ-5D-5L questionnaire, detailed in Table 3 [29]. These questionnaires have been validated for use in the breast cancer population [30,31]. All other secondary outcomes are listed in Table 3, with details of the methods of assessments, measurements variables, modes of assessment and methods of aggregation. Where available for the breast cancer population, minimum clinically important differences and psychometric properties are also noted in Table 3.

## Timeline

The SMART trial will recruit participants over an estimated two-year period which commenced on the 17th June 2024 and is estimated to end on the 17th June 2026. To allow for long term follow-up, planned data collection will continue for one-year following recruitment of the last participant, estimated to end in June 2027. At the time of submission of this paper, 56 participants had been recruited, with no data analysis, and no submission of publication of results. Results from the primary end point, 16 weeks following the baseline assessment, are expected by the end of 2026, and from the long-term follow-up by the end of 2027.

## Sample size calculation

Sample size calculations were conducted using the physical functioning scale in the EORTC BR-30 (continuous data) as the dependent variable. To detect a between group difference in this outcome of at least 8 (out of 100) (estimated minimal clinically important difference), [30] assuming a standard deviation of 20.5 (derived from published literature) with an $\alpha = 0.05$ and $1 - \beta = 0.8$, we will need to recruit 104 participants in the IG and 104 participants in the CG. To account for 20% drop-out, we plan to recruit a sample size of 130 participants in each group. Therefore, we aim to recruit 260 participants over 24 months.

## Data management and availability

Signed written informed consent forms will be maintained within locked cabinets in the physiotherapy department at RPH, only accessible to the research team. De-identified data will be collected and managed on REDCap. On analysis, data will be examined for outliers, implausible values, duplicates, and logic checks performed between variables.. Full data management procedures are detailed in the full protocol, *Supplementary Item 2*. As this is a protocol paper, no datasets were generated or analysed. The results will be published in a scientific journal. Upon study completion, the data generated from this study will be available from an online repository or from the corresponding author upon reasonable request, as far as possible under legislation.

**Table 3. Description, measurement variables, methods of assessment, and aggregation for secondary outcomes.**

| Outcome measure | Description of outcome and MCID* (*where MCID is available for the breast cancer cohort) | Measurement variable | Mode of assessment | Method of aggregation (if not continuous) | Psychometric properties (*where reported for the breast cancer population) |
|---|---|---|---|---|---|
| **Health-related quality-of-life** | | | | | |
| European Organisation for Research and Treatment of Cancer Core Questionnaire (EORTC C30) [27] | The C30 has both multi-item scales and single-item measures. These include five functional scales, three symptom scales, a global health status, and six single items. Each of the multi-item scales includes a different set of items – no item occurs in more than one scale. A high scale score represents a higher level of functioning/ quality of life or a high level of symptoms. The MCID for the varies for each sub-scale, depending on improvement or deterioration. Changes (out of 100) of 3.64 (improvement) and 4.28 (deterioration) have been estimated to be meaningful for the physical functioning subscale within a cancer population in Australia [32], which are lower than previously estimated scores of 8 [30]. | Global health status is transformed to a score out of 100. Each functional scale has 2–5 questions, which are each scored between 1–4. Linear transformation results in a functional scale score out of 100. Each symptom scale has 1–3 questions which are scored between 1–4. Linear transformation results in a symptom score out of 100. | Self-reported questionnaire sent via REDCap | N/A | Since the first version of the C30 was developed in 1993 it has undergone updates and validation in many international studies, has been used in more than 5000 studies worldwide and has been translated into 120 languages. [31] Recently, the content validity of the C30 has been reviewed, with the test-retest reliability for all scales with intraclass correlation coefficient (ICC) above 0.70. [31] To aid interpretation of the results, reference values for the EORTC C30 have been reported for adults with BC as well as thresholds for clinical importance [30,33]. |
| European Organisation for Research and Treatment of Cancer Breast Cancer Specific Questionnaire (EORTC BR45) [28] | The breast cancer module, the BR45 is used in conjunction with the C30. The BR45 includes nine multi-scale items and seven symptom scales/items to assess body image, sexual functioning, breast satisfaction, systemic therapy side effects, arm symptoms, breast symptoms, endocrine therapy symptoms, skin mycosis symptoms, endocrine sexual symptoms. In addition, single items assess sexual enjoyment, future perspective and being upset by hair loss. | Each functional scale has 1–4 questions, each scored between 1–4. Linear transformation results in a functional scale score out of 100. Each symptom scale has 1–10 questions which are scored between 1–4. Linear transformation results in a symptom score out of 100. | Self-reported questionnaire sent via REDCap | N/A | The EORTC BR45, which is the update from the prior BR23, has been validated for use in the breast cancer population [28]. The psychometric properties of the BR45 in a contemporary BC population reported the scale structure, internal consistency (for all multi-item scales Cronbach's alpha >0.7), test-retest reliability (ICC for all multi-item scales 0.68–0.83), convergent, discriminant, and clinical validity, and responsiveness to change [28]. Recently, the BR45 was updated to become the BR42 [34]. As data collection commenced prior to the update, the BR45 is used in this study. Reference values for the BR45 for adults with breast cancer are also available [30]. |

*(Continued)*

**Table 3.** (Continued)

| Outcome measure | Description of outcome and MCID* (*where MCID is available for the breast cancer cohort) | Measurement variable | Mode of assessment | Method of aggregation (if not continuous) | Psychometric properties (*where reported for the breast cancer population) |
|---|---|---|---|---|---|
| EuroQol 5D 5L (EQ-5D-5L) [29] | The EQ-5D-5L is a 5-item questionnaire and a visual analogue scale (EQ VAS). The questionnaire provides a simple descriptive profile of a respondent's health state. The EQ VAS provides an alternative way to elicit an individual's rating of their own overall current health.<br>When the descriptive system profile is linked to a 'value set', a single summary index value for health status is derived that can be used in economic evaluations of healthcare interventions. A value set provides values (weights) for each health state description according to the preferences of the general population of a country/region.<br>Index values are a major feature of the EQ-5D-5L instrument, facilitating the calculation of quality adjusted life years (QALYs) that are used to inform economic evaluations of healthcare interventions. The preferences of the general population of a country/region for different health states represent the societal perspective which, in general, is considered the preferred perspective in health economic analysis.<br>The minimum important difference (MID) for a cancer population for the EQ VAS is 5 points for both improvement and deterioration. [32] | EQ-5D-5L health states will be represented by a single summary number (index value), which reflects how good or bad a health state is according to the preferences of the general population of Australia. | Self-reported questionnaire sent via REDCap | N/A | The EQ-5D-5L has been validated for a broad range of health conditions, including cancer [29]. A systematic review of the psychometric properties of the EQ-5D-5L index score, with test-retest reliability reported in nine studies with moderate to excellent ICC (ICC > 0.7) [29]. |
| **Physical activity** | | | | | |
| **International Physical Activity Questionnaire (IPAQ)** [25] | Physical activity will be recorded using the IPAQ-SF which is a 7-item self-report questionnaire on vigorous and moderate activity and sedentary time. | Weekly exercise duration (in minutes) and frequency (number of days per week) will be recorded for:<br>-Moderate intensity aerobic exercise<br>-Vigorous intensity aerobic exercise<br>-Total (combined moderate and vigorous) aerobic exercise<br>-Resistance exercise | Verbal (phone based) interview with physiotherapist | In addition to the exercise minutes and frequency as continuous variables, a binary recording will be noted:<br>Were the guidelines of 150 minutes of moderate intensity aerobic exercise and two resistance sessions reached? (Yes/No) | There are no psychometric data for the IPAQ-SF in our study population, although there are reported values using the Spanish version of the IPAQ-SF in a Spanish cohort including breast cancer survivors [25].<br>In a comparison of the IPAQ with a wearable activity monitor, criterion validity was (N = 736) ρ = 0.3, 95% CI 0.23–0.36 [25].<br>Reliability (test-retest) performed one-week apart for first and second visits, and three days apart for second and third visits:<br>Pooled: Spearman's ρ = 0.76 (95% CI 0.73–0.77) [25]<br>Short form, last 7 days administered via telephone: ρ = 0.74 (N = 300) [25] |

*(Continued)*

**Table 3.** (Continued)

| Outcome measure | Description of outcome and MCID* (*where MCID is available for the breast cancer cohort) | Measurement variable | Mode of assessment | Method of aggregation (if not continuous) | Psychometric properties (*where reported for the breast cancer population) |
|---|---|---|---|---|---|
| **Physical assessment** | | | | | |
| **Body composition** | Assessed using bioimpedance spectroscopy, a non-invasive measure of body composition which provides estimates of tissue composition, fluid variables and a metabolic report [35]. Accurate recording of these variables requires the patient's weight be measured prior to assessment. | -Total body water (kg and % weight) -Extracellular fluid (kg and % weight) -Intracellular fluid (kg and % weight) -Hy-dex -Skeletal muscle mass (kg and % weight) -Fat mass (kg and % weight) -Fat free mass (kg and % weight) -Protein & minerals (kg and % weight) -Phase angle (degrees) -Active tissue mass (ATM) -Extracellular mass (ECM) | SOZO analyser (Impedimed, Brisbane, Queensland), during physical assessments. | | To the best of our knowledge, there is no psychometric data for the SOZO Bioimpedance spectroscopy in our patient cohort, however the SOZO is highly correlated with the L-Dex U400 (Impedimed Ltd, Brisbane, Australia) for the measurement of limb impedance [36]. Impedance ratio ($R0_{unaffected}$ : $R0_{at\text{-}risk}$) ($r = 0.921$, $p < 0.0001$) [37]. Coefficient of variance (CV) for impedance of $ECF = 0.6$, comparable to other BIS devices [36]. |
| **Six Minute Walk Test (6MWT) [38]** | Walking capacity will be measured using the six-minute walk test, performed according to standard guidelines. [38] The outcome measured is the distance in metres walked in 6 minutes (6MWD). Each participant will perform the test twice and the longest 6MWD will be recorded as the test result. The MCID for the 6MWD is reported as an increase in more than 30 metres from pre- to post-exercise intervention [39]. If a participant has Stage 4 breast cancer, the 6MWT is completed with modifications, providing participant is independent in their mobility. The 6MWT is performed as a pain limited/symptom limited test, as opposed to a maximal test. | 6MWD (metres) Pulse rate (beats per minute, maximum during test) Pulse rate (beats per minute, after 1 minute recovery) | Under physiotherapist instruction during physical assessments | | The reliability and validity of the 6MWT have been well reported for patient groups with cardiorespiratory diseases, but less commonly reported for those with cancer [40]. Keeping with other clinical populations, the test-retest reliability of the 6MWT for patient with cancer is reported as excellent, with ICC of 0.93 (95%CI: +0.86; +0.97) and a coefficient of variation of 3% [41]. For patients with newly diagnosed cancer who are undergoing pre-operative rehabilitation, the repeatability of the distance covered is excellent (ICC = 0.98, 95%CI: 0.92–0.99), but a learning effect is present with recommendation to perform two attempts of the 6MWT for this population.[40] |

*(Continued)*

**Table 3.** (Continued)

| Outcome measure | Description of outcome and MCID* (*where MCID is available for the breast cancer cohort) | Measurement variable | Mode of assessment | Method of aggregation (if not continuous) | Psychometric properties (*where reported for the breast cancer population) |
|---|---|---|---|---|---|
| **Grip strength** | Participants will be instructed to perform a maximal voluntary contraction. The participant will perform the test in sitting. Starting position will be shoulder in adduction, elbow flexion to 90 degrees, forearm and wrist in neutral position. The participant will be encouraged to perform a maximal squeeze of the dynamometer (*JAMAR Smart device*) for 3 seconds. The test will be repeated 5 times (on each hand) and the measure that is the highest, but within 10% of one other will be recorded as the test result.[42] The MCID for grip strength has been reported as in increase of at least 11 pounds (5 kg) pre- to post-exercise intervention.[39] If a participant has Stage 4 breast cancer with any bony metastases, grip strength assessments are completed with modifications, as a pain limited/symptom limited test, as opposed to a maximal test. | Left- and right-hand maximal test score (in kg). | Grip strength will be measured using a Jamar hand-held dynamometer (Surgical Synergies, SI Instruments, SA, Australia), under physiotherapist instruction during physical assessments | | Grip strength is a reliable indicator of overall muscle strength, and is also associated with overall mortality and HRQoL [43]. Grip strength measurement, using a handheld *JAMAR Smart* dynamometer is a valid and reliable measurement tool when used with adults in a hospital setting, with excellent intrasubject reliability (ICC 0.91–0.97), good intersession reliability (ICC 0.85–0.97), and correlates well with the gold standard tool, the JAMAR Hydraulic dynamometer (r = 0.68–0.98) [44]. |
| **Isometric muscle strength** | Participants will be instructed to perform a maximum voluntary isometric contraction of the middle deltoids, biceps brachii and quadriceps femoris. The participant will perform the test in sitting. Starting position will be shoulder in adduction and neutral external rotation, elbow flexion to 90 degrees, forearm and wrist in neutral position, hip and knee flexion to 90 degrees. The participant will be encouraged to 1) abduct their shoulder (for middle deltoids), 2) flex their elbow (for biceps brachii) and 3) to extend their knee (for quadriceps) with as much force as possible against the dynamometer for 3 seconds. The muscle meter will be stabilised with by the assessing therapist with the use of a nylon belt secured to the chair, to maximise reliability. The test will be repeated 5 times (on each limb) and the measure that is the highest, but within 10% of one other will be recorded as the test result [42]. If a participant has Stage 4 breast cancer with any bony metastases, the above isometric muscle tests are not completed. | Left- and right-hand maximal test score (in kg). | Isometric middle deltoids, biceps brachii and quadriceps strength will be assessed using a Lafayette Muscle Meter no. 01165 (SI Instruments, SA, Australia), under physiotherapist instruction during physical assessments | | The reliability and validity of isometric muscle strength has not been determined in patients following diagnosis with breast cancer, although hand-held dynamometry has been recommended for use in this population [45]. This has been investigated in adult patients following minor burn injury, using the protocol that has been adapted for this trial. The intraclass correlation coefficient and minimum detectable difference for maximum voluntary isometric contractions were: Middle deltoids: left – ICC 0.926 (95% CI 0.864, 0.956), MDD 5.15 kg; right – ICC 0.858 (95% CI 0.759, 0.924), MDD 6.59 kg. Biceps brachii: left – ICC 0.912 (95% CI 0.839, 0.954), MDD 7.65 kg; right – ICC 0.834 (95% CI 0.711, 0.911), MDD 9.82 kg. Quadriceps: left – ICC 0.870 (95% CI 0.767, 0.932), MDD 11.0 kg; right – ICC 0.837 (95% CI 0.711, 0.915), MDD 12.3 kg [42]. |

*(Continued)*

| Outcome measure | Description of outcome and MCID* (*where MCID is available for the breast cancer cohort) | Measurement variable | Mode of assessment | Method of aggregation (if not continuous) | Psychometric properties (*where reported for the breast cancer population) |
|---|---|---|---|---|---|
| **Healthcare Utilisation** | | | | | |
| **Institute for Medical Technology Assessment Medical Consumption Questionnaire (iMTA MCQ) [46]** | Healthcare utilisation will be recorded using the Medical Consumption Questionnaire (iMCQ), a non-disease specific questionnaire which aims to collect costs within the healthcare sector [46]. The language in this questionnaire has been adapted for suitability for an Australian population (*Supplementary Item 5*). | The total number of appointments with each type of health-care professional will be recorded. These will be grouped into outpatient appointments (hospital), outpatient appointments (non-hospital), emergency department presentations, inpatient admissions, home care and day care treatment. Descriptive data will be used in conjunction with hospital inpatient and outpatient booking data to enable clear healthcare utilisation journeys expected as part of usual care in breast cancer treatment. | Self-reported questionnaire sent via REDCap | N/A | To the best of our knowledge, the psychometric properties for this questionnaire have not been reported for the cancer or Australian adult population. However, this questionnaire is commonly used in the Netherlands to assess health utilisation for different clinical groups, including oncology populations [47]. |
| **Workplace absenteeism and presenteeism** | | | | | |
| **Institute for Medical Technology Assessment Productivity Cost Questionnaire (iMTA PCQ) [48]** | Workplace absenteeism and presenteeism will be assessed using the iMTA Productivity Cost Questionnaire (iPCQ), which aims to measure and subsequently value productivity costs [48]. The iPCQ has 3 modules: lost productivity at paid work due to absenteeism, lost productivity at paid work due to presenteeism and lost productivity at unpaid work. | Absenteeism will be recorded as the total number of absent days from work will be recorded. In addition, the total number of days with lost productivity will be recorded, with the average (mean) percentage of self-reported productivity on those days. | Self-reported questionnaire sent via REDCap | In addition to the total of days absent, absenteeism will be categorised as a short-term or long-term absence. Short-term absences are defined as those less than 4 weeks duration. Long-term absences are defined as those lasting longer than 4 consecutive weeks. | As productivity in terms of workplace absenteeism and presenteeism has not been reported among adults with BC in Australia, there are no reports of the psychometric properties for our cohort. However, it has been used for similar recent cohorts in the Netherlands [49]. Other populations including adults with musculoskeletal concerns in Norway, report good content validity (factor analysis revealed a 3-component solution accounting for 82% of the total variance) and test-retest reliability (ICC above 0.88) [50]. |

*(Continued)*

| Outcome measure | Description of outcome and MCID* (*where MCID is available for the breast cancer cohort) | Measurement variable | Mode of assessment | Method of aggregation (if not continuous) | Psychometric properties (*where reported for the breast cancer population) |
|---|---|---|---|---|---|
| **Psychological determinants and mechanisms of behavioural change** | | | | | |
| **Intention to exercise** [51] | Intention to exercise will be measured using 2 items (1 item each for intention to perform aerobic exercise and resistance exercise over the following week). Scores range from 1 to 7 for each item. Higher scores represent stronger intentions to engage in that exercise over the following week. | Score between 1 and 7 for each type of exercise (aerobic and resistance) | Self-reported questionnaire sent via REDCap | | A single item per behaviour to measure intention reduces participant burden and is suitable due to high correlations between items in lengthier measures [52]. |
| **Capability, Opportunity, Motivation – Behaviour (COM-B)** [53] | The COM-B questionnaire has 6 items (one for each subdomain of the COM-B model). Scores range from 1 to 11 for each item. Higher scores represent higher capability, opportunity, or motivation to engage in exercise. | Score between 1 and 11 for each subdomain | Self-reported questionnaire sent via REDCap | | This questionnaire was designed to be adaptable to different behaviours, and has demonstrated good to excellent reliability (ICCs .554−.833), and good discriminant validity, with acceptable pairwise correlations between items ($r$s < .85) [53]. |
| **Brief Self-Control Scale (BSCS)** [54] | The BSCS has 13 items answered on a 5-point Likert scale. Scores range from 1 to 5 with higher scores representing higher self-control. | Score from 1 to 5 | Self-reported questionnaire sent via REDCap | Average of item scores | This scale is commonly used to measure self-control in behaviour change studies and has shown good internal consistency ($\alpha = 0.96$) and test-retest reliability ($r = 0.89$, $p < 0.001$) [55]. |
| **Barrier Self-Efficacy Scale (BSES)** [55] | The BSES has 9 items answered on an 11-point Likert scale. Scores range from 0 to 10. Higher scores represent higher self-efficacy towards engaging in exercise when experiencing barriers. | Score from 0 to 10 | Self-reported questionnaire sent via REDCap | Average of item scores | This measure was developed to be used with adults with breast cancer and has shown good internal consistency ($\alpha = 0.96$) and test-retest reliability ($r = 0.89$, $p < 0.001$) [55]. |
| **Task Self Efficiency Scale (TSES)** [55] | The TSES has 4 items answered on an 11-point Likert scale. Scores range from 0 to 10. Higher scores represent higher self-efficacy towards engaging in exercise. | Score from 0 to 10 | Self-reported questionnaire sent via REDCap | Average of item scores | This measure was developed to be used with adults with breast cancer and has shown good internal consistency ($\alpha = 0.89$) and test-retest reliability ($r = 0.83$, $p < 0.001$) [55]. |
| **Habitual Tendencies Questionnaire (HTQ)** [56] | The HTQ is a 13-item questionnaire with 3 subscales: compulsivity, regularity, aversion to novelty. Each item is answered on a 7-point Likert scale – no item occurs in more than one scale. Scores range from 0 to 66 with higher scores representing more reliance on habitual behaviours. | Score between 0 and 66 | Self-reported questionnaire sent via REDCap | | This questionnaire measures individual's reliance on habits and has demonstrated good reliability ($\alpha = 0.76$) and good validity when compared to similar measures such as the Routine subscale of the Creature of Habit Scale [56]. |

*(Continued)*

**Table 3.** (Continued)

| Outcome measure | Description of outcome and MCID* (*where MCID is available for the breast cancer cohort) | Measurement variable | Mode of assessment | Method of aggregation (if not continuous) | Psychometric properties (*where reported for the breast cancer population) |
|---|---|---|---|---|---|
| **Self-Report Behavioural Automaticity Index (SRBAI)** [57] | The SRBAI has 4 items answered on a 7-point Likert scale. Scores range from 0 to 6 with higher scores representing stronger habit strength to engaging in exercise. | Score between 0 and 6 | Self-reported questionnaire sent via REDCap | Average of item scores | The SRBAI is an adaptable measure of the automaticity of behaviours and has demonstrated good internal consistency when adapted to assess physical activity behaviours ($\alpha = 0.74–0.89$) [58]. |
| **Mental Health** | | | | | |
| **Positive and Negative Affect Scale (PANAS)** [59] | The PANAS is a 20-item questionnaire with 2 subscales (positive and negative affect). Each of the items is answered on a 5-point Likert scale. Each subscale consists of 10 items – no item occurs in more than one scale. Each subscale ranges in score from 10 to 50. A high scale score represents higher levels of positive and negative affect. Lower positive affect scores and higher negative affect scores suggest higher levels of psychological distress. | Score between 10 and 50 for each subscale. | Self-reported questionnaire sent via REDCap | | The PANAS has been commonly used with adults with breast cancer, to assess positive and negative emotions, demonstrating high internal consistency ($\alpha = .76$) [60]. |
| **Work and Social Adjustment Scale (WSAS)** [61] | The WSAS has five items answered on a 9-point Likert scale. Scores range from 0 to 45. Higher scores represent higher impact of mental health on the ability to perform daily activities. | Score out of 45 | Self-reported questionnaire sent via REDCap | | This scale has demonstrated high internal consistency ($\alpha = 0.93$), and good validity across within adults with breast cancer [62]. |
| **Feasibility and Acceptability of the SMART program** | | | | | |
| **Feasibility/Acceptability Questionnaire** [63] | The Feasibility/Acceptability questionnaire has 8 items answered on a 5-point Likert scale. Each item is a statement about different perceptions of the program (e.g., annoying, easy to understand). Scores for each item range from 1 to 5 with higher scores representing stronger agreement with that statement. | Score between 1 and 5 for each statement | Self-reported questionnaire sent via REDCap | | To the best of our knowledge, the psychometric properties are not available for this outcome; however, it has been used frequently for process evaluations of interventions. |
| **Exit Interviews** | Semi-structured interview to identify experiences of the intervention components, such as preferences for intensity, duration, exercises, and behaviour change strategies, along with barriers to intervention participation. | Qualitative data | Verbal interview with research team member | | To the best of our knowledge, the psychometric properties are not available for this outcome. |
| **Chemotherapy Completion Rates** [64] **and Endocrine Therapy completion** | Chemotherapy plans and delivery will be extracted from patient medical records. Chemotherapy completion rates will be recorded as a mean relative dose intensity (RDI), which is the chemotherapy medication dose intensity received expressed as a percentage of the original planned dose intensity, accounting for both the dose and the number of weeks in the chemotherapy cycle [64]. An RDI of less than 85% is associated with worse survival outcomes [65]. Endocrine therapy prescription details and any deviations from the prescription plan (for example, dose reduction or discontinued use) will be extracted from the patient medical notes. In addition, the medication module from the above described iMTA MCQ will be used to ascertain if the medication is still being used at the long term follow up. | The RDI is a percentage score (0–100%) [64] Any dose adjustments or dose delays relating to endocrine or chemotherapy medication will be described using. | Patient medical records. | In addition to the percentage RDI value, the total number (n) of those with an RDI of less than 85% will be recorded. The total number (n) of those with any deviation from their endocrine therapy will be recorded. | To the best of our knowledge, the psychometric properties are not available for this outcome. |

## Analysis

To ensure the study statistician remains blinded to the study group, a member of the research team who is not involved in the analysis, will extract the group randomisation list and assign a generic name to each group (e.g., red and blue group). Patterns of missing data will be explored and multiple imputations considered for missing independent covariates where appropriate. [66,67].

The analyses for the primary research objective will be performed according to the intention-to-treat principle. For the primary outcome, HRQoL, with 4 assessment points, linear mixed models will be used to account for repeated measures with participant ID entered as a random effect. The primary time point of interest will be the end of the intervention phase. Group, time, and group-by-time interactions will be included as fixed effects in the regression model. The baseline measure of the outcome will be included as a covariate to adjust for baseline differences. The model will also be adjusted for randomisation variables and confounders, such as age, surgery type and weeks since surgery and/or chemotherapy pathway (neoadjuvant or adjuvant). Estimates of the effects will be reported with their corresponding 95% confidence intervals.

Between group differences for the secondary measures (peripheral muscle force-generating capacity, muscle mass, exercise tolerance) will be examined using the same method described for the primary outcome.

To understand any moderator effects, the listed covariates will be entered into a multivariable model using the outcomes listed above. In addition, adherence will be examined to see if it mediates the relationship with the outcomes, by calculating the indirect through the use of partial coefficients.

To address the research question investigating whether psychological determinants and mechanisms of behaviour change predict changes in physical activity levels in participants, bivariate correlations will first be conducted to explore associations between intention, self-efficacy, habit strength, and COM-B constructs (capability, opportunity, and motivation) with physical activity levels at each time-point. All relevant behavioural determinants will then be entered into a mixed-effect regression model, to examine if any behavioural determinants relate to physical activity levels.

The research question investigating whether physical activity is associated with psychological wellbeing will be assessed using correlations between physical activity levels, positive and negative affect data from the PANAS, and scores from the WSAS assessing impaired functioning due to mental health at each time-point. If appropriate, mixed-effects models will then be used to examine if physical activity relates to any of the psychological wellbeing variables.

Participants' perceptions of the SMART intervention will be explored using data from the feasibility/acceptability questionnaire [63]. Quantitative data will be assessed using descriptive statistics. Additionally, responses to the open-ended items from the questionnaire will be coded and analysed using conventional content analysis using an inductive coding approach [68]. Data from the exit interviews will also be analysed using qualitative methods (such as thematic analysis) to explore participants' reasons for engagement [or lack of engagement] [69].

Sub-group analysis will be performed for the HRQoL EORTC and EQ5D5L questionnaires for; i) those with Stage 4 breast cancer at the time of recruitment versus those with lower cancer stages, and ii) treatment regimens involving curative intent to treat versus palliative treatment. Model fit for regression models will be performed by examining residuals (normality and constant variance). For models with repeated measures random coefficients will also be examined for normality and constant variance. If model fit is poor, such as for skewed or bounded distributions then non-linear models will be explored using a gamma or inverse gamma generalised linear mixed model [70]. Any deviations from the plan will be described in the final manuscript.

## Trial governance, risk management and monitoring

A steering committee will meet monthly and oversee the running of the trial and administrative requirements. Any adverse events will be reported to the steering committee and recorded in the person's medical record and classified as minor (i.e., resolve without any need for a medical review) or non-minor (i.e., those that warranted a medical review). Examples of minor adverse events might include episodes of dizziness or nausea during exercise or muscle soreness and generalised

malaise on completion of an exercise session. Example of non-minor adverse events might include an injurious fall or cardiac event.

The trial intervention and assessments will be delivered by physiotherapists employed at RPH. Physiotherapists in Australia are trained in providing exercise prescription for a wide variety of health conditions. All trial physiotherapy staff are registered and fulfill the requirements of the Australian Health Professional Regulation Agency and have annual training to stay up to date with departmental emergency procedure training specific to gym-based exercise sessions and telehealth appointments run through the RPH.

A data and safety monitoring board (DSMB) has been established for this research project to review trial data every 6 months and for reporting of potential adverse events. The DSMB comprises three experienced researchers, independent to the clinical researchers actively working on the project. Any adverse events will be reported to the HREC within 48 hours, will be discussed at the DSMB meetings and reported to the treating medical team as appropriate.

## Discussion

This trial will investigate whether a more structured and supported exercise program provided via the SMART intervention will optimise exercise participation and improved health-related quality of life, and/or other secondary outcomes, compared to those receiving usual physiotherapy care alone. If successful, there will be justification to change routine clinical practice in a hospital setting. Heightened understanding of any variables that are associated with better outcomes will refine implementation of exercise interventions, so that physiotherapists or other exercise providers are able to optimise delivery of programs in a flexible way, with clear target goals to optimise outcomes through the breast cancer treatment journey. A greater understanding of the psychological determinants and mechanisms of change for exercise will guide interventions promoting exercise and the behavioural strategies underpinning them. As barriers to exercise are heightened for people undergoing a cancer journey, and people are at elevated risk of comorbidity following cancer treatment, exercise interventions are of particular importance for improving quality of life and related function and psychological outcomes in this population [14]. Findings will be shared through publications to inform usual care practices delivered by both the immediate care team and allied health professionals, harnessing the multidisciplinary care approach.

A major strength of this study was the co-design of the exercise intervention which took into consideration perspectives of several healthcare disciplines and, perhaps most importantly, the voices of people who have survived breast cancer. In addition, the exercise intervention was designed to be inclusive of both males and females, with any stage of breast cancer (including stage 4), who resided in either rural or metropolitan areas.

Limitations of this study involve the single centre design, which potentially will reduce the generalisability of the results, as usual care practices may differ between health services providers both within and outside Australia. Variations in usual care may reduce the generalisability of our estimate of the between-group differences. Although all follow-up assessments are completed by a researcher blinded to the study group, it was not possible to blind the participants or the physiotherapist delivering the intervention to their group allocation. This is a common shortcoming of studies that explore exercise interventions. Although the primary research question does not attempt to inform the mechanism of the interventions, our secondary analysis will allow us to explore factors that may modify the magnitude of between-group differences which will inform future service delivery. Overall, we feel there is high likelihood that the results will be clinically meaningful.

## Supporting information

**S1 File. Trial registration data.**
(DOCX)

**S2 File. Full protocol [as per version in ethics approval].**
(PDF)

**S3 File. SPIRIT checklist.**
(PDF)

**S4 File. Participant information and consent form [as per version in ethics approval].**
(PDF)

**S5 File. Healthcare utilisation questionnaire (revised).**
(PDF)

## Acknowledgments

In-kind support from the Physiotherapy Department at RPH has enabled the development and implementation of this study. One of the authors (ADG) is supported by an Australian Government Research Training Program Scholarship through Curtin University. We would also like to thank the staff in the Breast Clinic, Medical Oncology Clinics and Physiotherapy Department at Royal Perth Hospital, and the team at Sir Charles Gardner Hospital Radiation Oncology team for their ongoing involvement with this study.

## Author contributions

**Conceptualization:** Susan Stinton, Dale Edwick, Chloe Maxwell-Smith, Natasha Bear, Lauren Breen, Alejandro Dominguez Garcia, Elizabeth Dylke, Kate Edwards, Adam Lloyd, Sally Lynch, Niamh Moloney, Barbara Mullan, Ru-Wen Teh, Carol Watson, Kylie Hill.

**Funding acquisition:** Adam Lloyd, Carol Watson, Kylie Hill.

**Methodology:** Susan Stinton, Dale Edwick, Chloe Maxwell-Smith, Natasha Bear, Lauren Breen, Alejandro Dominguez Garcia, Elizabeth Dylke, Kate Edwards, Adam Lloyd, Sally Lynch, Niamh Moloney, Barbara Mullan, Ru-Wen Teh, Carol Watson, Kylie Hill.

**Project administration:** Susan Stinton, Sally Lynch, Carol Watson, Kylie Hill.

**Resources:** Susan Stinton, Chloe Maxwell-Smith, Lauren Breen, Alejandro Dominguez Garcia, Niamh Moloney, Barbara Mullan.

**Supervision:** Chloe Maxwell-Smith, Lauren Breen, Barbara Mullan, Kylie Hill.

**Writing – original draft:** Susan Stinton.

**Writing – review & editing:** Susan Stinton, Dale Edwick, Chloe Maxwell-Smith, Natasha Bear, Lauren Breen, Alejandro Dominguez Garcia, Elizabeth Dylke, Kate Edwards, Adam Lloyd, Sally Lynch, Niamh Moloney, Barbara Mullan, Ru-Wen Teh, Carol Watson, Kylie Hill.

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
