## [Decision Letter · Decision Letter 0]

29 Jul 2025

Dear Dr. Stinton,

Thank you for submitting your manuscript to PLOS ONE. After careful consideration, we feel that it has merit but does not fully meet PLOS ONE’s publication criteria as it currently stands. Therefore, we invite you to submit a revised version of the manuscript that addresses the points raised during the review process.

We look forward to receiving your revised manuscript.

Kind regards,

Mohammad Jobair Khan,

Academic Editor

PLOS ONE

Journal Requirements:

[This research is funded by the Vonesch Breast Cancer Rehabilition Grant (S01/2023), through the RPH Research Foundation. This funding was made possible from a gift in Memory of Felix Viktor Vonesch and Verena Vonesch.

Funding acquisition: KH, AL, CW

The RPH Research Foundation have provided peer review of this protocol design but do not play a role in the conduct of the study; collection, management, analysis, and interpretation of the data; preparation, review or approval of the manuscript; and decision to submit the manuscript for publication.

https://www.rphresearchfoundation.org.au/].

We note that one or more of the authors is affiliated with the funding organization, indicating the funder may have had some role in the design, data collection, analysis or preparation of your manuscript for publication; in other words, the funder played an indirect role through the participation of the co-authors. If the funding organization did not play a role in the study design, data collection and analysis, decision to publish, or preparation of the manuscript and only provided financial support in the form of authors' salaries and/or research materials, please do the following:

1. Review your statements relating to the author contributions, and ensure you have specifically and accurately indicated the role(s) that these authors had in your study. These amendments should be made in the online form.

2. Confirm in your cover letter that you agree with the following statement, and we will change the online submission form on your behalf:

“The funder provided support in the form of salaries for authors [insert relevant initials], but did not have any additional role in the study design, data collection and analysis, decision to publish, or preparation of the manuscript. The specific roles of these authors are articulated in the ‘author contributions’ section.

[This research is funded through the Vonesch Breast Cancer Rehabilitation Research Grant which was made in memory of Felix Viktor Vonesch and Verena Vonesh through the RPH Research Foundation. Additional in-kind support from the Physiotherapy Department at RPH has been received to enable the development and implementation of this study. One of the authors (ADG) is supported by an Australian Government Research Training Program Scholarship through Curtin University. We would like to thank the staff in the Breast Clinic, Medical Oncology Clinics and Physiotherapy Department at Royal Perth Hospital, and the team at Sir Charles Gardner Hospital Radiation Oncology team for their ongoing support.]

[This research is funded by the Vonesch Breast Cancer Rehabilition Grant (S01/2023), through the RPH Research Foundation. This funding was made possible from a gift in Memory of Felix Viktor Vonesch and Verena Vonesch.

Funding acquisition: KH, AL, CW

The RPH Research Foundation have provided peer review of this protocol design but do not play a role in the conduct of the study; collection, management, analysis, and interpretation of the data; preparation, review or approval of the manuscript; and decision to submit the manuscript for publication.

https://www.rphresearchfoundation.org.au/]

5. In the online submission form, you indicated that [This is a protocol paper, no datasets were generated or analysed. Upon study completion, the data generated from this study will be available from an online repository or from the corresponding author upon reasonable request, as far as possible under legislation. Full data management procedures are detailed in the full protocol, Supplementary Item 2.].

6. One of the noted authors is a group or consortium [SMART Study Team]. In addition to naming the author group, please list the individual authors and affiliations within this group in the acknowledgments section of your manuscript. Please also indicate clearly a lead author for this group along with a contact email address.

Additional Editor Comments:

I have reviewed the protocol with keen interest. The manuscript is very engaging, but a few comments need to be addressed before it can be published. Here are my comments:

Title: In the title, mention study settings and design, i.e., Self-administered dual-task training reduces balance deficits and falls among community-dwelling older adults: a multicentre parallel-group randomised controlled trial with economic evaluation protocol

Introduction:

For references, please update all citations to include only those from the last five years in the introduction. Use a current cohort of references that is no older than five years.

Additionally, please avoid using multiple references unnecessarily; try to limit to one or, at most, two references.

In lines 85 to 88, explain why you chose the SMART intervention. What are the differences from other interventions, like Otego exercise and a self-administered dual task?

In lines 90 to 96, rewrite the following standard manuscript.

In lines 98 to 105, explain why you want to “explore” (line 98) …., “investigate” (line 101) …and “assess” (line 103)…. From “explore, investigate, and assess,” who will benefit and how?

Method

Please recheck the manuscript with the SPIRIT 2013 CHECKLIST and use subheadings as appropriate for better clarification of the methods.

In 172 to 333, explains outcome measurements. Please use Table 2 throughout the explanation of measurements instead of using almost all measurements.

In line 287, Table 2 explains the different outcomes. Please use references for Outcome measures, Descriptions, and Measurement variables.

In line 363, what is the meaning of the logic check? Please explain.

In lines 364-365: Please explain with appropriate reference why multiple imputations are considered.

In line 372, what covariates will be used in the analysis with reference? How will you determine the covariates?

Line 373: Please list all confounders you wish to include in the analysis.

In the methods section, there is a repetition of some information. Please remove the repetition.

Please follow the referencing in the main text following Plos One standard (i.e., https://journals.plos.org/plosone/article?id=10.1371/journal.pone.0327947)

In line 441: how suddenly a hypothesis appears. I have not seen any hypothesis mentioned in the introduction or methods. Please mention the hypothesis with an adequate explanation in the introduction and other parts as appropriate.

In lines 451 to 453: please use reference for the following statement “As barriers to exercise are heightened for people undergoing a cancer journey, and people are at elevated risk of comorbidity following cancer treatment”.

458-460: Why do you think a single center is a limitation and reduces generalisability? Your study has several additional limitations, including the blinding of the assessor, statistician, and intervention provider, as well as the limitation of cost-saving analysis, and it also does not inform the mechanism of interventions. Please discuss these in detail, referencing them as appropriate.

451-454: Please use reference for following sentences: “As barriers to exercise are heightened for people undergoing a cancer journey, and people are at elevated risk of comorbidity following cancer treatment, exercise interventions are of particular importance for improving quality of life and related function and psychological outcomes in this population.”

Please add a paragraph on the strength of the study.

Line 459-460: Instead of the word “Confidence”, use a scientific word.

Please correct the page orders.

Reviewers' comments:

Reviewer's Responses to Questions

**Comments to the Author**

1. Does the manuscript provide a valid rationale for the proposed study, with clearly identified and justified research questions?

Reviewer #1: Yes

Reviewer #2: Yes

2. Is the protocol technically sound and planned in a manner that will lead to a meaningful outcome and allow testing the stated hypotheses?

Reviewer #1: Yes

Reviewer #2: Yes

3. Is the methodology feasible and described in sufficient detail to allow the work to be replicable?

Reviewer #1: Yes

Reviewer #2: Yes

4. Have the authors described where all data underlying the findings will be made available when the study is complete?

Reviewer #1: Yes

Reviewer #2: Yes

5. Is the manuscript presented in an intelligible fashion and written in standard English?

Reviewer #1: Yes

Reviewer #2: Yes

You may also provide optional suggestions and comments to authors that they might find helpful in planning their study.

Reviewer #1: This is a well written protocol with an appropriate sample size calculation included and a good description of the intended analyses to be conducted. I have only two points to make.

1. A very large amount of data is to be collected on each patient so I imagine the great logistical difficulties associated. I wonder therefore if some (say 5 to 10%) of the less important variables might be dropped?

2. Note a SD quoted to 2 decimal places seems a little unrealistic. I would have assumed SD = 21 with n then increasing from 105 to110 as a consequence! Clearly, the authors choice of the total sample size of N = 260 is fine.

I wish them good luck with the trial.

Reviewer #2: Thank you very much for the opportunity to review this manuscript. Overall, the logical flow of the manuscript is clear, and the methodology of the clinical randomized controlled trial is appropriate. After a thorough reading, I have provided several comments aimed at enhancing the quality of the work.

1. Page 12-28: Please provide the minimal clinically important difference (MCID) for each outcome measure in the table.

2. Page 32: Given that the linear mixed effects model can appropriately handle missing data, the application of additional imputation methods may not be required. You may wish to consider whether the use of imputation is necessary in this context.

3. Page 31-32: As the linear mixed effects model will be employed in the data analysis, please also consider assessing the normality of the residuals. Please put it in the method part.

4. Page 31: For the sample size calculation, please place the reference citation [33] immediately following the estimated minimal clinically important difference. This would improve both the accuracy and clarity of the citation.

5. Other format of manuscript, please follow the requirements of PLOS One.

**Do you want your identity to be public for this peer review?** For information about this choice, including consent withdrawal, please see our Privacy Policy

Reviewer #1: No

Reviewer #2: **Yes:** Dr. Hong Pan, Department of Rehabilitation Science, The Hong Kong Polytechnic University

---

## [Author Response · Author response to Decision Letter 1]

11 Sep 2025

To the Editor,

Thank you for the opportunity to re-submit our manuscript with revisions, which were due by the 12th September 2025.

Your comments and suggestions have been addressed in the attached "Response to reviewers" letter.

Please let us know if you would like us to address anything further.

Kind regards,

Susan Stinton

---

## [Editor Report · Decision Letter 1]

12 Oct 2025

Dear Dr. Susan Stinton,

Thank you for submitting your manuscript to PLOS ONE. After careful consideration, we feel that it has merit but does not fully meet PLOS ONE’s publication criteria as it currently stands. Therefore, we invite you to submit a revised version of the manuscript that addresses the points raised during the review process.

We look forward to receiving your revised manuscript.

Kind regards,

Mohammad Jobair Khan, MPH

Academic Editor

PLOS ONE

Journal Requirements:

Additional Editor Comments:

Title

- In lines 1 and 2, please include the study design—specifically, "single-blinded multicenter randomized controlled trial"—in the title: “Evaluating the effect of the SMART intervention in people with recently diagnosed breast cancer: protocol and statistical analysis plan.”

Abstract

- Line 40: The term "usual care" in the statement “Both groups will receive usual care” is unclear. Please clarify what is meant by usual care in this context.

- Line 41: Please specify what SMART stands for. Ensure that all acronyms are defined in full the first time they appear in the manuscript.

- Line 42: The phrase “Theory-informed” should reference the specific theory guiding the intervention.

- Line 47: The outcome will be assessed before randomization and at 8, 16, and 52 weeks. The word “later” is confusing; please specify the exact time points for outcome assessment.

- Line 49: The statement “Chemotherapy and endocrine therapy completion rates will be recorded” requires clarification. Please explain the relevance of recording chemotherapy and endocrine therapy completion rates in relation to the exercise intervention. Additionally, indicate whether this will serve as a secondary outcome and provide justification.

- Line 32: While this study presents a new intervention, there is insufficient discussion about it. Please specify and describe the latest interventions being considered.

- Line 55: The Materials section lacks a discussion of the factors included. Please specify what these factors are.

Introduction

- Please update references to include those published within the last five years.

- Line 59: Consider using a more precise alternative to the word “dramatically,” or remove it.

- Lines 58–64: Only one side effect is mentioned, but line 64 refers to “these unwanted side effects.” Please revise line 64 to: “Strategies are needed to mitigate these unwanted side effects.”

- Line 71: Reference 14 is not a meta-analysis, and no forest plot or explanation of meta-analysis is provided. Please replace with an appropriate reference.

- Lines 75–83: Include information on endocrine treatment, as the trial will include this aspect.

- Lines 90–96: Please remove this section from the introduction.

- Please explain the need to explore variables that influence the magnitude of any change in Health-Related Quality of Life (HRQoL) and clearly state your objective. Similarly, for psychological determinants, mental health, and other factors (lines 100–105), this paragraph should follow line 88, not appear as a separate paragraph.

Methods

- It is suggested to include a table listing all resistance and aerobic exercises to be performed in the trial. Also, mention the phases of exercise (i.e., warm-up, main exercise, and cool-down) with allocated minutes for each phase.

- Line 287: Please mention the psychometric properties for all outcome measures reported in Table 2.

- Line 291: Please summarize all physical assessments in a single paragraph.

- Line 344: While the primary outcome is Health-Related Quality of Life (HRQoL), it was not used in the sample size calculation. It is highly recommended to calculate the sample size based on primary pilot data for HRQoL.

- Please include a paragraph about blinding, specifying who will be blinded and how blinding will be maintained.

- The sentence “a will be examined for outliers, implausible values, duplicates, and logic checks performed between variables” should be included in the "Data Management and Availability" section.

- Line 364: Please explain why multiple imputation was chosen, with adequate references.

- Specify the data distribution for which the model fit will be chosen, clarify which mixed model will be used, and provide justification for this choice, including references.

- Lines 380–382: Please provide references for these statements.

- Line 386: Specify the outcomes being referred to.

Discussion

- Line 459: If generalizability is impacted, outline the statistical or other procedures or steps that will be taken to resolve or minimize the generalizability issue.

---

## [Author Response · Author response to Decision Letter 2]

24 Nov 2025

To the Editor,

Thank you for the ongoing support to revise our manuscript for PLOS One. Our replies to your suggestions from your letter dated the 13th October 2025 are in the attached document, titled "Response to reviewers letter 19112025".

Kind regards,

Susan Stinton

---

## [Decision Letter · Decision Letter 2]

8 Jan 2026

Evaluating the SMART intervention in people with recently diagnosed breast cancer who are being treated at a public tertiary hospital in Australia: protocol and statistical analysis plan for a single-blinded, single centre randomised controlled trial

PONE-D-25-25500R2

Dear Dr. Stinton,

We’re pleased to inform you that your manuscript has been judged scientifically suitable for publication and will be formally accepted for publication once it meets all outstanding technical requirements.

Kind regards,

Mario Lopes, Ph.D

Academic Editor

PLOS One

Additional Editor Comments (optional):

Congratulations to the authors on the excellent work and substantial effort invested in the manuscript. The reviewers have accepted it for publication in its current form.

Reviewers' comments:

Reviewer's Responses to Questions

**Comments to the Author**

1. Does the manuscript provide a valid rationale for the proposed study, with clearly identified and justified research questions?

Reviewer #1: Yes

Reviewer #2: Yes

2. Is the protocol technically sound and planned in a manner that will lead to a meaningful outcome and allow testing the stated hypotheses?

Reviewer #1: Yes

Reviewer #2: Yes

3. Is the methodology feasible and described in sufficient detail to allow the work to be replicable?

Reviewer #1: Yes

Reviewer #2: Yes

4. Have the authors described where all data underlying the findings will be made available when the study is complete?

Reviewer #1: Yes

Reviewer #2: Yes

5. Is the manuscript presented in an intelligible fashion and written in standard English?

Reviewer #1: Yes

Reviewer #2: Yes

You may also provide optional suggestions and comments to authors that they might find helpful in planning their study.

Reviewer #1: Recommedation

Accept

I have nothing to add to my earlier statistical review when I recommended acceptance of this article.

Reviewer #2: The author revised my suggestion, and I am very satisfied. I recommend accepting it. No need any revision.

**Do you want your identity to be public for this peer review?** For information about this choice, including consent withdrawal, please see our Privacy Policy

Reviewer #1: No

Reviewer #2: **Yes:** Dr. Hong Pan

---

## [Editor Report · Acceptance letter]

PONE-D-25-25500R2

PLOS One

Dear Dr. Stinton,

I'm pleased to inform you that your manuscript has been deemed suitable for publication in PLOS One. Congratulations! Your manuscript is now being handed over to our production team.

Kind regards,

on behalf of

Prof. Mario Lopes

Academic Editor

PLOS One